# Occurrence and Dietary Exposure Assessment to Enniatin B through Consumption of Cereal-Based Products in Spain and the Catalonia Region

**DOI:** 10.3390/toxins15010024

**Published:** 2022-12-29

**Authors:** Jose A. Gallardo, Sonia Marín, Antonio J. Ramos, German Cano-Sancho, Vicente Sanchis

**Affiliations:** 1Technology, Engineering and Science of Food Department, AGROTECNIO-CERCA Center, University of Lleida, 25198 Lleida, Spain; 2Oniris, INRAE, LABERCA, 44300 Nantes, France

**Keywords:** enniatin B, deoxynivalenol, exposure assessment, cereal-based food, Fusarium mycotoxins

## Abstract

Enniatin B (ENNB) is a mycotoxin produced by moulds from the Fusarium genera and its toxic effects are still not fully elucidated, hence a safe reference exposure value has not been established yet. ENNB is the most prevalent emerging mycotoxin and is widely found in cereal-based products, nevertheless, there are no comprehensive exposure assessment studies. For that reason, the aim of this study was to characterise the occurrence of ENNB and estimate the exposure of the Spanish and Catalan populations. A total of 347 cereal-based products were collected in 2019 and were analysed using liquid chromatography-tandem mass spectrometry. Consumption data were obtained from the national food consumption surveys (ENALIA) and a regional survey conducted in Catalonia. The global exposure was estimated using deterministic and probabilistic methods. The results showed a high occurrence of close to 100% in all foodstuffs, with a range from 6 to 269 µg/kg, and a strong correlation with the levels of deoxynivalenol. Children aged one–nine years were the most exposed, showing mean estimates in the range 308–324 ng/kg bw/day and 95th percentiles 697–781 ng/kg bw/day. This study stresses the need for further toxicological data to establish reference doses and conclude formal risk assessment, accounting for the co-occurrence with deoxynivalenol.

## 1. Introduction

Mycotoxins are chemical substances synthesised by some filamentous fungi in response to stress conditions [1]. The presence of mycotoxins in the food chain is mainly due to mould growth in cereal crops in the field, but poor storage conditions can favor mycotoxins’ presence as well [2]. To date, about three hundred mycotoxins have been identified and almost thirty of them may cause adverse effects on human health [3]. Due to the toxic nature of mycotoxins, the European Commission has established a maximum value in food products for aflatoxins, ochratoxin A, patulin, deoxynivalenol (DON), zearalenone, fumonisins B1 and B2, and toxins T-2 and HT-2 [4]. Nonetheless, the European Commission has not established a maximum level for other emerging mycotoxins, like enniatins (ENN). In 2014, the European Food Safety Authority (EFSA) CONTAM Panel considered that there were not enough data to establish a Tolerable Daily Intake (TDI) and/or an Acute Reference Dose for the sum of ENN mycotoxins [5].

The toxicity of ENN is related to their ionophoric properties by forming complexes with alkali metal ions and increasing the cationic permeability of membranes [6]. Furthermore, effects on cell viability and apoptosis induction were observed by the high permeation from blood to the brain across the blood-brain barrier [7]. Recent evidence suggests that Enniatin B (ENNB) can produce Reactive Oxygen Species (ROS) targeting the intestinal barrier [8], affecting the embryonic development in mice [9], and impairing the reproductive function of cattle in vitro [10]. An interactive effect of ENNB with DON has been observed in piglets, which in combination led to decreased weight gain and diversity of gut microbiome [11].

Among the four ENN isoforms (A, A1, B, and B1), ENNB is the most relevant metabolite in terms of occurrence, which is produced by Fusarium species and is usually found in cereals, flour, and their derived food products [12,13,14,15,16]. Noteworthy, since moulds of the Fusarium genus can produce several mycotoxins [12], it is common to find multiple Fusarium mycotoxins co-occurring in cereal-based foodstuffs [17]. The characterization of their co-occurrence is further stressed by the reported joint effects of ENNB with other Fusarium mycotoxins, with the potential of enhancing their toxic effects [18,19].

The main exposure pathway to mycotoxins among the general human population is through diet, hence dietary exposure modelling becomes a suitable tool to estimate global exposure. Despite several comprehensive studies assessing the exposure of populations to main mycotoxins [3,20,21,22,23], to the best of our knowledge, there is no study focusing on ENNB. Thus, the present study aims at characterising the occurrence of ENNB in cereal-based food commonly consumed in Spain and estimating the dietary exposure of the general population, with a special focus on the region of Catalonia.

## 2. Results and Discussion

### 2.1. Occurrence of Enniatin B

Among the 23 mycotoxins analysed, only ENNB and DON (reported elsewhere [24]) were detected above the limit of detection (LOD: 2 µg/kg) in all analysed composites, whereas in baby food it was found only in 28% of samples (Table 1). Bread rusks were the most contaminated food item followed by tin bread (mean of 178.68 and 163.32 µg/kg, respectively), whereas baby food showed the lowest contamination values (mean of 6.71 µg/kg). Sweet bakery products also showed lower contamination values than the other bakery products (between 11 and 24 µg/kg for muffins, other cookies, cookies, and cakes). Conversely, the lower contamination levels of wholegrain bread than white or tin bread were unexpected because commonly the most contaminated part of grain is removed during the processing of white bread [25,26]. Considering the fact that the baking processes may equally affect the stability of ENN, we attributed these slight differences to the different origins of flours with different background contamination levels. Previous studies conducted in Spain showed similar values in pasta, within a range from 0.50 to 122.13 µg/kg in dry pasta and 0.50 g to 33.13 µg/kg in fresh pasta [27], and a range of 0.23 to 10.54 µg/kg in pizza dough [28], a product that we did not analyse but has a resemblance to bread. Infant formula was also analysed; ENNB_1_, ENNA, and ENNA_1_ but not ENNB were detected [29].

Some studies conducted in other countries showed, in general, lower occurrence values than those found in Spain; for instance, a mean of 25 µg/kg in pasta from Italy [30]. Lower values than the present study were also reported in the Netherlands, with a mean of 16 µg/kg in bread, 28 µg/kg in breakfast cereals, or 8.7 µg/kg in pasta [31]. Consistent values with our findings were reported in biscuits and cookies from the Netherlands (mean 15 µg/kg). In Tunisia, a range of 0.05 to 1.1 µg/kg in white bread, 0.39 to 10.7 µg/kg in wholegrain bread, and 0.05 to 1.6 µg/kg in biscuits, were found [32]. Likewise, a range between 0.37 to 1.05 µg/kg was detected in cereal-based products from China [33]. In the case of food for children, in the Netherlands 17 µg/kg were found in bread, 16 µg/kg in breakfast cereals, and 58 µg/kg in rye and maize products [31]. In Austria, a range from <0.4 to 40 µg/kg was reported in infant foods [34]. In Tunisia, ranges from 22 to 93 µg/kg in breakfast cereals, and 6 to 8 µg/kg in infant cereals were found [35].

We found that the concentration of ENNB was strongly and positively correlated with the levels of DON (Figure 1), previously analysed in the same samples [24], with a Pearson correlation coefficient of 0.627 (*p* < 0.0001). Several samples showed concentrations systematically above the trend line, corresponding to wholegrain bread. Globally, the concentrations of ENNB exceeded those of DON, showing an average ratio ENNB:DON of 1.8 (range 0.3–6.8). The type of food item contributed to explaining the variability of ratios of concentrations, for instance, the mean ratio in wholegrain bread was 0.4, whereas in cookies, 2.3. These differences may be a consequence of the differential effects of food processing among both mycotoxins [25]. These results further support the need for a mixed risk assessment of ENNB combined with DON, especially in light of potential synergistic effects reported in toxicological studies [18].

### 2.2. Consumption of Cereal-Based Foods

Consumption patterns of cereal-based food among the population of Spain and the Catalonia region were described elsewhere [24]. Briefly, the most consumed cereal-based products among the Spanish young population (under 18 years) was white bread, followed by cookies and pasta, whereas the remainder of the foodstuffs were consumed occasionally (Table 2). Among adults, bread and pasta accounted for the most consumed food products in terms of percentage of consumers and average consumption, showing some slight differences among Spain and the Catalonia region, and Spain (Appendix A).

### 2.3. Exposure Assessment of Enniatin B among Adult Population

Exposure estimates among the adult population from Spain and the Catalonia region are summarized in Table 3. Median estimates were in the range between 105 and 174 ng/kg bw/day, whereas the 95th percentile ranged between 202 and 452 ng/kg bw/day. Slight differences were observed between age groups and studied regions; the group aged 18–40 years old showed the highest estimates, especially for Spain, irrespective of the exposure assessment method (probabilistic vs. deterministic). The main contributor to ENNB exposure for the Spanish adult population (Appendix A) was white bread (53.6–62%) followed by pasta (17.2–20.1%), and tin bread (10.6–19.5%). For the Catalonian, pasta was the main contributor (33.2–36.5%), white bread (15.8–25.5%), and breakfast cereals (13.2–18.6%).

Dietary exposure to ENNB has seldom been addressed in previous studies, or even rarely considered in total diet studies [36]. The few available studies (summarized in Table 4) showed that our results were lower than those estimates provided by the EFSA Contam Panel for the European countries and cross-countries estimates (mean 420–1820 ng/kg bw/day and P95 910–3280 ng/kg bw/day). Nonetheless, our figures are in the range of those previously estimated in the Netherlands with a median of 47.3 ng/kg bw/day to 54.4 ng/kg bw/day, and a P95 of 113 ng/kg bw/day to 124 ng/kg bw/day [37]. The estimates in Algeria using contamination data on raw wheat showed exposure estimates of 13,960 ng/kg bw/day [17], and the highest value was from Austria, with 40,000 ng/kg bw/day [34], about 400 times higher than the ones found in our work.

### 2.4. Exposure Assessment of Enniatin B among Infant and Adolescent Population

The exposure estimates for the young population under 18 years old are shown in Table 5. Catalonian exposure was calculated only with the infant food consumption of infants up to three years old, for which we considered two scenarios, a Lower Bound (LB) scenario where censored contamination data was substituted by 0, and an Upper Bound (UB) scenario where censored data was substituted by a LOD of 2 µg/kg. The estimation of both scenarios showed an exposure below 100 ng/kg bw/day, except for P95 and P99 of UB, with estimates of 129.1 ng/kg bw/day for P95 and 166.6 ng/kg bw/day for P99 only with infant formula consumption. For the estimation of the minor population from Spain, we followed the same methodology as for the adult population and with the same food groups. The exposure estimates were strongly determined by the age group, with children aged one–nine years being the most exposed with median and 95th percentile estimates in the ranges 264–266 and 697–710 ng/kg bw/day, respectively. Little differences were observed between exposure assessment methods. The population between 0 and 11 months old showed the lowest exposure estimation, which could be related to the consumption data in Table 2, where this age group’s consumption was almost limited to white bread and cookies, whereas the other age groups showed a wider consumption pattern (e.g., 50% of pasta consumers).

The highest contributor to ENNB exposure was white bread (45.5–54.1%), crackers (34.0–39.5%), and the third one was basic cookies for 0–11 month old infants (18.6%), cookies (3.4–6.4%) for children between three and nine years old, and pasta (3.1%) for those aged 10 to 17 year (Appendix A). As found for adults, the high consumption of white bread played an important role in exposure contribution, and pasta contributed as well but had less relevance in minors than in adults. The fate of ENNB during pasta cooking is still not clear. Some authors suggested that ENNB suffers from thermal degradation [27], but results from other authors showed 80–100% of ENNB retention during cooking due to hydrophobic characteristics [39]. Further studies will be necessary to clarify these differences.

The exposure results from our study were lower than other previously published studies (Table 5). In Europe, the mean range of exposure to ENNB ranged from 47.3 to 40,000 ng/kg bw/day [5,34,37], whereas the mean range of our study ranged from 28.32 to 266.0 ng/kg bw/day (including all methods). These large differences suggest a need for further ENNB studies to refine the exposure assessments. In addition, some differences between the existing studies can account for that variability, like the number of volunteers, the number of samples, the contamination values, the different consumption patterns among the cultures, and time-frames. For instance, among non-EU countries, China showed the lowest mean exposure to ENNB in the infant population (2 ng/kg bw/day) [38], whereas similar exposure estimations to our study were found in Tunisia [35].

## 3. Conclusions

To the best of our knowledge, this is the first comprehensive exposure assessment study of the emergent mycotoxin ENNB. The results support the widespread presence of this emergent mycotoxin in cereal-based food highly consumed in Spain. The median levels ranged between 10 µg/kg (muffins) to 193 µg/kg (bread rusks). A high correlation between ENNB and DON concentrations was found, with mean ENNB:DON ratios of 1.8 (range 0.3–6.8). Children aged one–nine years were the most exposed group with mean and 95th percentile estimates in the ranges 308–324 ng/kg bw/day and 95th percentile 697–781 ng/kg bw/day, respectively. The main contributor to the exposure to ENNB for all ages and regions was white bread and pasta, except for babies, which was white bread. These results also highlight the relevance of adult food (e.g., bread and pasta) on global exposure estimates among infants transitioning their diet at early ages. Globally, these results support the need for further toxicological data of ENNB alone and in combination with DON to conduct formal risk characterisation of this emergent toxin.

## 4. Materials and Methods

### 4.1. Food Sampling and Preparation

The overall methodology for sampling has been described elsewhere [24]. Between 24 April 2019 and 14 May 2019, a total of 220 different cereal-based food samples were collected from different establishments, according to the type of sample. From bakeries among the four Catalan provinces (Lleida, Tarragona, Barcelona, and Girona), a total of 127 samples of white bread and wholemeal bread samples were collected (Appendix A). Breakfast cereals, pasta, bakery products like tin bread (white tin bread), bread rusks, and crackers, and sweet bakery products like cookies (which were basic cookies), other cookies (which include chocolate cookies, cream cookies, butter cookies, tea cookies, and digestive cookies), muffins and cakes, were collected from three different types of distributors: hypermarkets (group 1), supermarkets (group 2) and local distributors (group 3). The 32 baby food samples were collected by brand, not by location.

Firstly, the samples were dried at 40 °C before their homogenization and milling (except bread and baby food) which was performed in a Thermomix 3000 obtained from Vorwerk (Wuppertal, Germany). Once milled, samples were mixed by groups in analytical composites, obtaining a total of 27 composites (three composites per product). For bread, 4 composites were obtained by combining the individual samples from each Catalonian region. The 32 individual samples of baby food were analysed separately.

### 4.2. Chemical Analysis of Mycotoxins

Deepoxi-deoxynivalenol (DOM) in a concentration of 50.3 µg/mL was used as an internal standard, purchased from Sigma Aldrich (St. Louis, MO, USA). Composites were spiked with 50 ppb of this standard. For the extraction, 2 g from each composite (or individual sample in the case of baby food) were placed in a Falcon tube with 18 mL of the extraction solvent, which was a mixture of acetonitrile (analytical reagent grade 99.99%) purchased from Fischer Scientific (Loughborough, UK), bi-distilled water, and formic acid (analytical reagent grade 98–100%) obtained from Fischer Scientific (Leicestershire, UK) in a proportion of 60:37:3, respectively. Falcon tubes were placed in a minitron orbital shaker from INFORS AG CH-4103 HT (Bottmingen, Switzerland) at 200 rpm for 30 min. After that, a previously prepared mixture of 4 g of magnesium sulphate anhydrous (technical grade 96%) obtained from PanReac Applichem (Darmstadt, Germany) with 1 g of sodium chloride (reagent grade ≥ 99.5%) purchased from Fischer Scientific (Loughborough, UK) was added and manually shaken for 60 s and shaken again in the orbital shaker for 30 min. After that time, the mixture was centrifuged at 3000 rpm for 10 min in a centrifuge Universal 320R purchased from Hettich (Tuttlingen, Germany). 7 mL of the organic fraction was taken and placed in a glass tube to evaporate the solvent at 40 °C under a gentle stream of nitrogen.

For the greasy samples, the mixture of the sample and the extraction solvent was centrifuged at 5000 rpm for 5 min, after that, 10 mL of n-hexane 96% obtained from Scharlab (Sentmenat, Spain) was added and shaken again in the orbital shaker at 200 rpm for 30 min. Once shaken, the mixture was centrifuged one more time at 5000 rpm for 5 min, and the hexane was discarded by decantation. The rest of the procedure followed the same method as for the other samples.

The above-mentioned extraction methods were performed as requested by the laboratory of the Centre of Excellence in Mycotoxicology and Public Health, Department of Bioanalysis, Faculty of Pharmaceutical Sciences, Ghent University (Ghent, Belgium), which also performed the determination and quantification of multi-mycotoxins in the evaporated extracts.

### 4.3. UPLC-MS/MS Analysis Procedure

The liquid chromatography-tandem mass multi-detection method allowed the quantification of 23 toxins as previously reported in the reference method [40]. The sample analysis was performed in a Waters^®^ Acquity UPLC system coupled to a Quattro XEVO TQS mass spectrometer (Waters^®^, Manchester, UK). The software used for data acquisition and processing was MassLynx™ version 4.1 and QuanLynx^®^ version 4.1 (Waters^®^, Manchester, UK). Separation of analytes was carried out by an HSS T-3 column (2.1 × 100 mm, 1.8 μm) (Waters^®^, Manchester, UK). Two different mobile phases were used, mobile phase A (water/methanol/acetic acid; 94/5/1, *v*/*v*/*v*) and mobile phase B (methanol/water/acetic acid; 97/2/1, *v*/*v*/*v*), both buffered with 5 mM ammonium acetate and were set at a flow rate of 0.3 mL/min. The total run time was 28 min. Five µL of sample was injected into the UPLC system.

The ESI-source was operated both in negative and positive mode (ESI− and ESI+), and parameters were optimized and programmed for all measurements as follows: source and desolvation temperatures were 150 °C and 200 °C, respectively; the capillary voltage was 30 kV and nitrogen was applied as spray gas. The argon collision gas pressure was 9 × 10^−6^ bar, the cone gas, and desolvation gas flow were 150 and 550 L/h, respectively. The LOD was 2 μg/kg and method validation showed apparent recoveries between 97 and 108%, well within the range 80–110% established by Commission Decision 2002/657/EC.

### 4.4. Cereal-Based Food Consumption Data

#### 4.4.1. National-Based Consumption Data

Consumption data of general Spanish population were obtained from the National Dietary Survey on the Child and Adolescent Population in Spain (ENALIA) and the Spanish National dietary survey in adults, elderly, and pregnant women (ENALIA2) conducted by the Spanish Agency for Consumer Affairs, Food Safety and Nutrition (AESAN) [41]. The design, protocol, and methodology of the ENALIA studies, following the EFSA “EU Menu” guidance recommendations, have been described in other studies as well [42].

ENALIA is a dietary recall conducted between 2012 and 2014 that was based on two non-consecutive 1-day food diaries for children aged 6 months to 10 years, and two 24-h dietary recalls for 11 to 17-year-old children and adolescents, separated by at least 14 days to ensure that information best resembled usual dietary intake.

On the other hand, ENALIA2 consisted of a dietary survey conducted between 2013 and 2015 with a sample of 900 adults/elderly and 133 pregnant women. The food intake was evaluated with two non-consecutive days (at least 14 days in between) 24-h recalls and a Food Propensity Questionnaire [41].

#### 4.4.2. Regional-Based Consumption Data

A self-administered online food frequency questionnaire (FFQ) was conducted for the most consumed cereal-based foods in the region of Catalonia, which may represent a source of *Fusarium* mycotoxins exposure, to obtain individual consumption data. The semi-quantitative FFQ described in other studies as well [43] was completed between 8 May 2019 and 1 July 2019 by 1564 adult Catalan people divided into four age ranges: 18–25 years old (N = 562), 26–40 years old (N = 307), 41–60 years old (N = 401) and elder people above 60 years old (N = 295).

The consumption data were normalised by dividing the consumption of the food by the individual body weight (bw), and the result was expressed as g of food/kg bw/day. The previously published cereal-based infant food consumption data was used to provide specific exposure estimates in the region [43].

### 4.5. Exposure Assessment

Deterministic and probabilistic methods were used in the present work to combine the contamination data with the consumption data.

Using Equation (1), we combined the summary contamination data and the consumption data assuming independence between consumption (*C_π,j_*) and contamination (*T_j_*) to perform the deterministic method [43].
(1)$${\hat{E}}_{\pi_{0}}=\overset{p}{\underset{j=1}{\sum }}{\bar{C}}_{\pi_{0},\;j}\cdot \bar{T_{j}}$$
where: ${\hat{E}}_{\pi_{0}}$ is the normalised exposure of sample population *π*_0_;$\;{\bar{C}}_{\pi_{0},\;j}$ is the arithmetic mean of normalised consumption of food *j* by population π_0_ and $\bar{T_{j}}$ is the arithmetic mean of contamination of food *j*.

The probabilistic method used in this work consisted of a semi-parametric method (SP) previously described in other studies [43,44].

Firstly, with graphical methods and the Anderson-Darling statistics support, we attempted to fit the probability density functions (*pdf*) on consumption datasets. Next, with EnviroPRA v1.0 package R software [45], we did 10,000 iterations to sample the normalised consumption of the food *j* from its respective *pdf* and integrate it with the mean level of mycotoxin contamination of the food *j* using again the Equation (1) to estimate the individual exposure profiles.

We considered the fixed arithmetic mean contamination in the probabilistic models to estimate the individual common exposures and related chronic risk instead of drawing acute exposure profiles. In both probabilistic methods, 10,000 iterations were considered.

## Figures and Tables

**Figure 1 toxins-15-00024-f001:**
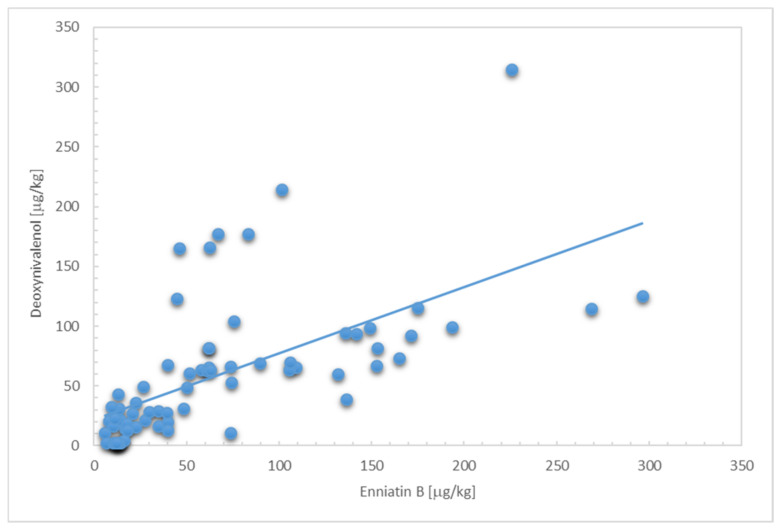
Scatter plot depicting the associations between concentration levels of enniatin B and deoxynivalenol in cereal-based foodstuffs, excluding infant food. Nine samples with non-detected levels of deoxynivalenol were assigned the limit of detection of 2 µg/kg.

**Table 1 toxins-15-00024-t001:** Occurrence of enniatin B (µg/kg) in analysed cereal-based foodstuff.

	*n*	DF	Mean	Median	Min	Max
White bread	4	100	103.5	105.1	62.16	142.0
Wholemeal bread	4	100	74.46	75.11	46.30	101.3
Tin bread	3	100	163.3	175.1	74.70	225.4
Bread rusks	3	100	178.7	193.6	73.75	268.7
Crackers	3	100	69.15	48.32	22.67	136.5
Cookies	3	100	20.60	13.73	13.24	34.82
Other cookies	3	100	19.48	11.89	6.66	39.88
Muffins	3	100	11.68	10.36	7.90	16.79
Cakes	3	100	23.42	27.62	12.97	29.67
Breakfast cereals	3	100	113.8	148.9	39.65	153.0
Pasta	3	100	110.6	39.62	47.19	152.5
Baby food *	32	28	6.71	4.65	2.53	16.79

* Individual contamination values of baby food analysed by brands. Abbreviations: *n*, number of analysed composites. DF, detection frequency (% of samples above the limit of detection at 2 µg/kg).

**Table 2 toxins-15-00024-t002:** Normalised consumption (g/kg bw/day) of cereal-based products for the population under 18 years old from ENALIA survey in Spain. Mean levels and standard deviation (SD) were computed considering only consumers identified as percentage of consumers (% cons).

Food	Age	N	%	Mean	SD
	Group	Total	Cons		
White bread	0–11 months	182	43%	1.13	0.82
	12–35 months	333	79%	1.81	1.21
	3–9 years old	589	95%	1.59	0.88
	10–17 years old	627	97%	1.15	0.61
Wholegrain bread	0–11 months	182	1%	0.89	0.41
	12–35 months	333	2%	3.64	3.14
	3–9 years old	589	6%	1.44	0.77
	10–17 years old	627	9%	0.98	0.60
Bread rusks	0–11 months	-	-	-	-
	12–35 months	333	4%	0.78	0.74
	3–9 years old	589	5%	0.56	0.47
	10–17 years old	627	3%	0.33	0.15
Crackers	0–11 months	182	5%	0.31	0.11
	12–35 months	333	12%	0.98	2.20
	3–9 years old	589	5%	0.92	0.96
	10–17 years old	627	5%	0.55	0.64
Cookies	0–11 months	182	81%	1.04	0.55
	12–35 months	333	74%	1.37	0.83
	3–9 years old	589	48%	1.27	0.78
	10–17 years old	627	34%	0.79	0.50
Other cookies	0–11 months	-	-	-	-
	12–35 months	333	1%	1.54	0.95
	3–9 years old	589	3%	1.62	1.10
	10–17 years old	627	2%	1.30	1.12
Muffins	0–11 months	182	1%	3.33	-
	12–35 months	333	10%	2.55	1.06
	3–9 years old	589	16%	1.73	0.79
	10–17 years old	627	16%	1.08	0.59
Cake	0–11 months	-	-	-	-
	12–35 months	333	0.30%	7.89	-
	3–9 years old	589	1%	3.91	1.93
	10–17 years old	627	2%	2.06	0.83
Breakfast cereals	0–11 months	182	2%	2.92	3.18
	12–35 months	333	3%	1.75	1.35
	3–9 years old	589	9%	1.13	0.95
	10–17 years old	627	8%	0.69	0.49
Pasta	0–11 months	182	9%	2.28	1.60
	12–35 months	333	49%	2.38	1.70
	3–9 years old	589	58%	1.76	1.32
	10–17 years old	627	59%	1.10	0.92

**Table 3 toxins-15-00024-t003:** Exposure to enniatin B (ng/kg bw/day) among the adult population. The estimates were calculated with the deterministic and probabilistic method through consumption of cereal-based food from the Catalonia region and Spain considering only consumers using the upper bound scenario for left-censored data.

Method	Region	Age Group	Median	Mean	SD	Min	P75	P95	P99
Deterministic	Catalonia	18–40	130.6	138.1	72.33	7.14	178.8	269.2	350.0
		41–60	109.1	114.9	59.86	7.22	143.2	222.5	299.7
		>60	105.3	111.4	54.34	4.79	146.1	202.8	259.7
	Spain	18–40	172.6	196.3	132.8	6.77	260.2	423.7	565.2
		41–60	152.2	170.1	100.7	4.69	219.4	338.1	453.4
		>60	144.5	154.2	93.91	1.79	213.7	338.6	396.9
Probabilistic	Catalonia	18–40	131.0	138.5	68.10	0.00	180.3	261.5	327.4
		41–60	109.5	116.6	61.95	0.00	153.7	228.3	290.1
		>60	105.4	112.0	55.45	0.00	146.2	213.0	263.8
	Spain	18–40	174.0	199.0	137.6	0.00	267.6	452.4	651.5
		41–60	148.2	174.6	123.0	0.00	234.0	405.2	583.3
		>60	133.6	161.0	125.8	0.00	214.2	391.5	606.0

Abbreviations: SD, standard deviation.

**Table 4 toxins-15-00024-t004:** Summary of exposure assessments to enniatin B among adult population reported in previous studies.

Country	Contamination Data	Group	Consumption Data	Mean Estimated Exposure (ng/kg bw/Day)		Reference
				Mean	P95	
Summary European countries		Infants		420–650	2070–2120	[5]
		Toddlers		1130–1820	2170–2930	
		Other children		870–1800	1510–3210	
		Adolescents		550–1070	1070–2030	
		Adults		470–1560	910–3280	
		Elderly		610–1710	980–3120	
		Very Elderly		660–1500	1040–2100	
Netherlands		2–6 years old	*n* = 5515	1131–144 (P50)	240–262	[37]
	*n* = 1617	7–69 years old	*n* = 3819	47.3–54.4 (P50)	113–124	
Austria	*n* = 59	6 months old		40,000		[34]
Tunisia	*n* = 115	Adults		3.95		[32]
		Children		7.7		
	*n* = 117	Children		36–357		[35]
Algeria	*n* = 30 (wheat)	Adults		13.96		[17]
China	*n* = 820	<3 years old	*n* = 20,172	20	50 (P97.5)	[38]

**Table 5 toxins-15-00024-t005:** Exposure to enniatin B (ng/kg bw/day) among the population under 18 years. The estimates were calculated with the deterministic and probabilistic method through the consumption of cereal-based food from the Catalonia region and Spain considering only consumers.

Method	Region	Age Group	*n*	Median	Mean	SD	Minimum	P75	P95	P99
Deterministic LB	Catalonia	0–36 months	133	28.32	29.45	22.15	0.79	41.68	73.27	94.57
Deterministic UB		0–36 months	133	49.89	51.87	39.01	1.4	73.40	129.1	166.6
Deterministic	Spain	0–11 months	182	45.96	100.8	133.6	5.72	120.5	328.0	622.2
		12–35 months	333	263.9	324.2	262.1	5.42	458.6	797.5	1181.3
		3–9 years	589	264.3	313.0	199.6	10.47	404.8	710.2	970.5
		10–17 years	627	182.2	213.0	131.8	11.63	270.8	447.2	673.2
Probabilistic	Spain	0–11 months	10,000	46.21	94.82	123.0	0.00	132.6	327.6	590.0
		12–35 months	10,000	254.5	308.4	234.5	0.00	422.3	780.8	1092.2
		3–9 years	10,000	266.0	310.8	201.1	0.00	401.0	697.1	1032.9
		10–17 years	10,000	185.7	224.2	154.2	0.00	283.18	511.7	813.5

Abbreviations: SD, standard deviation; LB, lower-bound; UB, upper-bound.

## Data Availability

The raw contamination data is available upon request. Spanish food consumption data is property and managed by the Spanish Agency for Consumer Affairs Food Safety and Nutrition, Spanish Minister of Consumer Affairs and Social Welfare.

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
