# Peer review of "Occurrence and Dietary Exposure Assessment to Enniatin B through Consumption of Cereal-Based Products in Spain and the Catalonia Region"

_toxins, 2022, doi:10.3390/toxins15010024_

Round 1

Reviewer 1 Report

This paper characterize the occurrence of ENNB and estimate the exposure of Spanish and Catalan population by evaluating the concentration of mycotoxins in 347 cereal-based products. The result showed that children aged 1-9 years were the most exposed ones, with the mean estimates in the range 308-324 ng/kg bw/day. So far, there is no clear safe reference exposure value for Enniatin B, the result is meaningful, and can give suggestion for the exposure risk assessment emergent mycotoxin ENNB.  Some suggestion list below:

 1.Key Contribution: deoxunivalenol, there is a spelling mistake.

 2. Results and discussion

Line 72-73, the lower contamination levels of 72 wholegrain bread than white or tin bread was unexpected because commonly the most 73 contaminated part of grain are removed during the processing of white bread, the authors should explain the possible reason for this result.

3. the English language should be improved.

Author Response

This paper characterize the occurrence of ENNB and estimate the exposure of Spanish and Catalan population by evaluating the concentration of mycotoxins in 347 cereal-based products. The result showed that children aged 1-9 years were the most exposed ones, with the mean estimates in the range 308-324 ng/kg bw/day. So far, there is no clear safe reference exposure value for Enniatin B, the result is meaningful, and can give suggestion for the exposure risk assessment emergent mycotoxin ENNB.  Some suggestion list below:

 Authors’ comments: Thank you for your time and comments, we have considered all of them in this new version

 1.Key Contribution: deoxunivalenol, there is a spelling mistake.

 Authors’ comments: Thanks, we have updated

  1. Results and discussion

Line 72-73, the lower contamination levels of 72 wholegrain bread than white or tin bread was unexpected because commonly the most 73 contaminated part of grain are removed during the processing of white bread, the authors should explain the possible reason for this result.

 Authors’ comments:

  1. the English language should be improved.

 Authors’ comments: We have thoroughly revised and improved the entire text

Reviewer 2 Report

The paper is well-written and well-organized. The quality of the research is good and demonstrates understanding of all aspects of the issue.

In the text there were some problems with the link to the tables ( line 67,, line 93, line112, line 121, line 137, line 148, line 162)

I believe that followed published papers should be considered in the article: https://doi.org/10.3390/toxins13100728

https://doi.org/10.1371/journal.pone.0197406

https://doi.org/10.3390/app122010566

https://doi.org/10.1080/19440049.2016.1228126

https://doi.org/10.1021/acs.jafc.0c00411

Author Response

Authors’ comments: Thank you for your time and comments, we have considered all of them in this new version

Reviewer 3 Report

I kindly ask the authors to better explain the analytical technique used for identification and quantification of results, in the following way:

Type of analytical instrument on which the analysis was performed, year of manufacture and manufacturer

Specify recording conditions on the instrument

Name of the used column (Type and manufacturer)

Specify the method's limit of quantification (LOQ)

Recovery (%)

I also suggest adding a chromatogram of the used standard and one representative chromatogram with positive results to the Supplemental materials.

Author Response

Authors’ comments: Thank you for the suggestion, we have added the details in the methods. We were unable to get the chromatogram from the external laboratory who conducted the analysis within this short time.